# Prevalence of multimorbidity in Brazilian individuals: A population-based study

**Ana Daniela Silva da Silveira** [1]*, **Jonas Eduardo Monteiro dos Santos** [2], **Marianna de Camargo Cancela**[3], **Dyego Leandro Bezerra de Souza**[4]

**1** School of Dentistry, Federal University of Pará-UFPA, Belém, Pará, Brazil, **2** Department of Quantitative Methods in Health, Sérgio Arouca National School of Health, Rio de Janeiro, Rio de Janeiro, Brazil, **3** Division of Population Research, National Cancer Institute, Division of Population Research, Rio de Janeiro, Brazil, **4** Department of Public Health, Federal University of Rio Grande do Norte, Natal, Rio Grande do Norte, Brazil

\* anadanielass@gmail.com

## Abstract

This study aimed to estimate the prevalence of multimorbidity in Brazilian individuals and its association with sociodemographic and lifestyle factors. This cross-sectional study used data from the National Health Survey conducted in 2019 with 88,531 Brazilian adults Multi-morbidity is the presence of two or more non-communicable chronic diseases. Associated factors were assessed by calculating the prevalence ratio (PR) obtained using Poisson regression with robust variance. Multimorbidity was estimated in 29.9% (95%CI: 29.33% to 30.48%) of Brazilian individuals. In the multiple models, the prevalence was high in women (PR: 1.37; 95%CI: 1.32 to 1.42), individuals over 60 years (PR: 4.26; 95%CI: 3.87 to 4.69), non-employed (PR: 1.20; 95%CI: 1.15 to 1.26), individuals with obesity (PR: 1.49; 95%CI: 1.43 to 1.56), and smokers (PR: 1.24; 95%CI: 1.19 to 1.29). This study identified a high prevalence of multimorbidity and its association with sociodemographic and lifestyle factors. The monitoring of these outcomes may support the development of policies and services.

## Introduction

Multimorbidity is the presence of two or more chronic non-communicable diseases (NCDs) [1–5]. The literature shows that interactions among NCDs are more harmful to the quality of life than NCDs that occur separately [5,6].

The increasing prevalence of NCDs is a worldwide phenomenon resulting from the westernization of lifestyle processes and population aging. Obesity, unhealthy eating habits, physical inactivity, and smoking are etiological factors contributing to NCDs (e.g., diabetes, high blood pressure, and cancer) [1,6–9]. Data from the 2013 National Health Survey (NHS) involving 60,202 individuals showed that 23.6% of the population has multimorbidity, which was mainly associated with sex (women), aging, low education level, smoking, and obesity [1].

Multimorbidity is a growing condition worldwide and is associated with worse health outcomes, increased use of services, and higher costs for healthcare systems. Although there is evidence that multimorbidity is prevalent worldwide, studies regarding its prevalence in Brazil and its association with other determinants are still in the beginning. Therefore, this research

**Funding:** The author(s) received no specific funding for this work.

**Competing interests:** The authors have declared that no competing interests exist.

aims to fill this gap in the literature and provide an updated estimate of the prevalence of multimorbidity in the Brazilian population. The results of this research can be useful in guiding public health policies and identifying vulnerable populations that need specific interventions.

The prevalence of multimorbidity in Brazilian adults directly impacts their quality of life and public health costs. In this sense, its monitoring is needed to create integral public policies focused on social and lifestyle factors [1,6].

Therefore, this study aimed to estimate the prevalence of multimorbidity in Brazilian adults over 18 years in 2019 and its association with sociodemographic and lifestyle factors.

## Methodology

This cross-sectional study analyzed data from the National Health Survey 2019 (NHS 2019) conducted by the Brazilian Ministry of Health and the Brazilian Institute of Geography and Statistics (IBGE). Data were public and had unrestricted access [10].

The NHS 2019 had a complex sampling designed in multiple stages, and sample selection was based on the IBGE master sample. The multi-stage sampling used in the NHS involved the selection of population elements in three stages of aggregation: (i) with stratification of the primary sampling units (census tracts), (ii) selection of households and (iii) household resident aged 15 or over. The selection of each stage was carried out randomly and systematically, and the sample size was calculated to ensure the representativeness of the results for the Brazilian population aged 15 years and over, residing in urban and rural areas of all states of Brazil [10,11].

The original sample of this research was analyzed using complex statistics, considering stratification effects in the estimation of indicators and measures: primary sampling units, stratum and weight. In other words, the NHS 2019 was representative at the national and state level for the Brazilian population with the characteristics described above [10,11].

The NHS 2019 assessed sociodemographic and lifestyle factors of Brazilian families in three parts: (i) the household; (ii) all household residents, focusing on sociodemographic and health information; and (iii) the selected resident, investigating information such as lifestyle, NCDs, and violence. Anthropometric measurements were also performed (sub-sample). A total of 108,525 households were visited across the country, and 94,114 interviews were performed. The non-response rate was 6.4% [10].

We analyzed 88,531 responses about the lifestyle of Brazilian individuals corresponding to the responses of people over 18 years old. Multimorbidity was defined by the presence of two or more NCDs. In this study, the prevalence of NCDs was self-reported and based on the response to the question "Has a doctor ever diagnosed you with (name of the disease)?".

The following NCDs were considered fourteen chronic diseases, according to Fig 1.

Prevalence estimates (%) and respective confidence intervals (95%) were calculated according to to sex, age group, ethnic group (self-reported), education level (years of schooling), occupation (employed and no employed), physical activity, body mass index (BMI), alcohol consumption, smoking, residence area (urban or rural), Brazilian macro-regions (north, northeast, southeast, south, and midwest), private health insurance, and medical care in the last year [10].

Alcohol consumption was classified as abstaining, moderate (two drinks a day for men and one drink a day for women), or excessive (five drinks or more in a day for men and four drinks or more in a day for women). Smoking was classified as never smoked, ex-smoker, or smoker [12]. For BMI, we used the World Health Organization (WHO) classification: underweight (BMI < 18.5 kg/m$^2$), normal weight (BMI between 18.5 and 24.5 kg/m$^2$), overweight (BMI between 25 and 29.9 kg/m$^2$), and obesity (BMI $\geq$ 30 kg/m$^2$). Physical activity was classified as

| Chronic disease | |
|---|---|
| 1 | Systemic arterial hypertension |
| 2 | Issues with vertebral spine (chronic back or neck pain. sciatic pain. lumbago. issues with vertebrae or disks) |
| 3 | Chronic kidney failure |
| 4 | Hypercholesterolemia |
| 5 | Heart problems (infarction, angina, or heart failure) Health issues (infarction, angina or heart failure) |
| 6 | Cerebrovascular accident or stroke |
| 7 | Asthma or asthmatic bronchitis |
| 8 | Arthritis or rheumatism |
| 9 | Chronic back pain |
| 10 | Work-related musculoskeletal disorder |
| 11 | Depression |
| 12 | Mental illnesses such as schizophrenia, bipolar disorder, psychosis or obsessive-compulsive disorder |
| 13 | Lung issues, such as pulmonary emphysema, chronic bronchitis or chronic obstructive pulmonary disease |
| 14 | Cancer |

**Fig 1. NCDs in the adult Brazilian population—NHS, Brazil, 2013.**

none, not enough, or enough, according to the total number of minutes per week dedicated to physical exercise, sports, leisure, efforts at work, commuting from work, or other displacements and household chores that involved physical effort. The WHO classifies the level of physical activity as moderate (75 to 150 minutes) or intense (150 to 300 minutes) [13].

Poisson regression with robust adjustment of variance verified associations between multimorbidity and independent variables; data regarding the prevalence ratio (PR) of multimorbidity and 95% confidence intervals were also estimated. Statistically significant models ($p \leq 0.05$) were entered into the multiple model. The Wald test assessed the significance of the models with p-value $<0.05$. All analyzes were performed using the statistical package STATA (Stata Corp. Inc. TX, USA, version 14) and adjusted for complex sample design.

The National Research Ethics Commission approved the NHS 2019 in August 2019 (registry no. 3,529,376). This study was conducted using secondary data available from publicly and unrestrictedly national surveys (www.ibge.gov.br) [10,11].

## Results

This paper analyzed 88,531 individuals data corresponding to an estimated 159,171,311 individuals. A total of 53.16% individuals were women; 21.61% had advanced age ($> 60$ years); and 12.59% were smokers. The prevalence of multimorbidity was 29.9% with an estimated frequency of 47,598,091 Brazilian individuals (Table 1).

The most frequent NCDs in both sexes were systemic arterial hypertension, chronic back pain, and hypercholesterolemia. A high prevalence of multimorbidity was observed among

**Table 1. Estimated prevalence of multimorbidity according to sociodemographic characteristics, social determinants, lifestyle factors, and use of health services.**

| Variables | N | % (95%CI[a]) |
|---|---|---|
| **Sex** | | |
| Men | 16,894,158 | 22.66 (21.94 to 23.39) |
| Women | 30,703,933 | 36.29 (35.48 to 37.10) |
| **Age group** | | |
| 18 to 29 | 3,271,881 | 9.30 (8.51 to 10.15) |
| 30 to 39 | 5,425,932 | 16.24 (15.30 to 17.22) |
| 40 to 49 | 8,048,579 | 27.82 (26.54 to 29.14) |
| 50 to 59 | 11,344,470 | 41.63 (40.07 to 43.21) |
| 60+ | 19,507,228 | 56.71 (55.61 to 57.80) |
| **Ethnic group** | | |
| White | 22,493,190 | 32.67 (31.79 to 33.56) |
| Non-white | 25,104,901 | 27.80 (27.09 to 25.51) |
| **Education level** | | |
| 0 to 3 | 1,348,680 | 43.79 (40.60 to 47.02) |
| 4 to 7 | 21,082,639 | 37.81 (36.84 to 38.79) |
| 8 to 10 | 12,641,951 | 22.12 (21.30 to 22.96) |
| 11 or more | 9,250,973 | 25.68 (24.54 to 26.85) |
| **Employed** | | |
| Yes | 21,986,724 | 22.55 (21.86 to 23.24) |
| No | 25,611,367 | 41.54 (40.63 to 42.47) |
| **Physical activity** | | |
| None | 13,078,640 | 38.10 (36.90 to 39.31) |
| Not enough | 8,307,504 | 32.85 (31.45 to 34.28) |
| Enough | 26,211,947 | 26.33 (25.66 to 27.02) |
| **BMI** | | |
| Normal weight | 14,418,484 | 23.11 (22.33to 23.91) |
| Underweight | 833,051 | 24.43 (20.86 to 28.38) |
| Overweight | 18,438,237 | 31.64 (30.72 to 32.58) |
| Obesity | 13,674,174 | 40.85 (39.56 to 42.15) |
| **Alcohol consumption** | | |
| Abstaining | 31,376,263 | 34.09 (33.33 to 34.85) |
| Moderate | 10,800,988 | 27.03 (26.02 to 28.06) |
| Excessive | 5,420,840 | 19.96 (18.90 to 21.06) |
| **Smoking** | | |
| Never smoked | 24,748,533 | 25.57 (24.89 to 26.27) |
| Ex-smoker | 17,141,204 | 40.46 (39.30 to 41.63) |
| Smoker | 5,708,354 | 28.49 (27.04 to 30.00) |
| **Residence area** | | |
| Urban | 41,789,954 | 30.47 (29.82 to 31.11) |
| Rural | 5,808,137 | 26.40 (25.30 to 27.54) |
| **Brazilian region** | | |
| North | 2,695,262 | 21.57 (20.52 to 22.66) |
| Northeast | 11,260,993 | 26.74 (25.88 to 27.63) |
| Southeast | 22,512,241 | 32.56 (31.47 to 33.66) |
| South | 7,826,332 | 33.48 (32.19 to 34.81) |
| Midwest | 3,303,263 | 27.42 (26.12 to 28.76) |

*(Continued)*

**Table 1.** (Continued)

| Variables | N | % (95%CIª) |
|---|---|---|
| **Private health insurance** | | |
| Yes | 15,579,447 | 33.02 (31.94 to 34.12) |
| No | 32,018,644 | 28.59 (27.97 to 29.22) |
| **Medicines used in the last year** | | |
| Yes | 44,652,577 | 34.78 (34.11 to 35.45) |
| No | 2,945,515 | 9.57 (8.87 to 10.31) |

ªCI: Confidence interval

ᵇ*p*: Wald test.

women (36.29%; 95%CI: 35.48 to 37.10) and individuals over 60 years (56.71%; 95%CI: 55.61 to 57.80), when compared to prevalence in men and younger individuals. The results showed that age exerted a dose-response effect on the occurrence of multimorbidity (Fig 2).

Regarding sociodemographic factors, a high prevalence of multimorbidity was found in individuals with low education level (over three years of schooling) (43.79%; 95%CI: 40.60 to 47.02) and non-employed (41.54%; 95%CI: 40.63 to 42.47), when compared to the other categories.

Regarding lifestyle factors, ex-smokers had the highest estimated values for multimorbidity (40.46%; 95%CI: 39.30 to 41.63) than individuals who never smoked. People classified as obese had higher prevalence values (40.85%; 95%CI: 39.56 to 42.15) when compared to other BMI categories. The prevalence of multimorbidity according to independent variables is described in Table 2. All variables were included in the multiple regression model.

The multiple regression showed that the prevalence of multimorbidity among women (PR: 1.37; 95%CI: 1.32 to 1.42) was 37% higher than in men, regardless of other variables. In the adjusted analysis, the prevalence significantly increased with age: multimorbidity was 3.26-fold higher in individuals over 60 years (PR: 4.26; 95%CI: 3.87 to 4.69) than in individuals aged between 18 and 29 years.

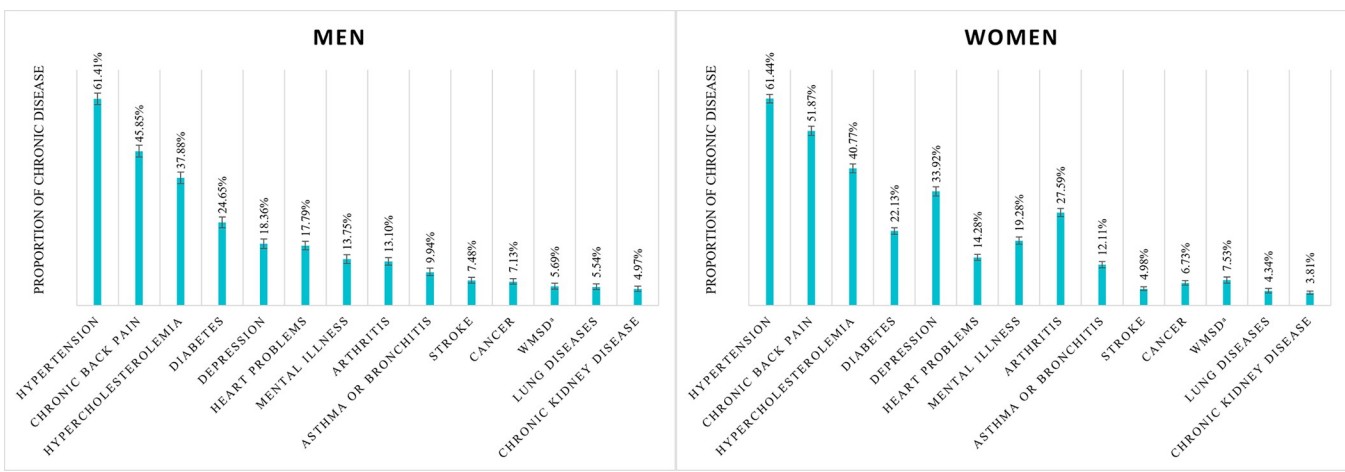

ªWMSD: Work-related musculoskeletal disorders

**Fig 2. Distribution of NCDs in individuals with multimorbidity according to sex.**

**Table 2.** Prevalence of multimorbidity in Brazilian individuals according to lifestyle and social factors.

| Variables | PR[a] non-adjusted | 95%CI[b] | p[c] | PR adjusted | 95%CI | p |
|---|---|---|---|---|---|---|
| **Sex** | | | | | | |
| Men | 1.00 | | | 1.00 | | |
| Women | 1.60 | 1.54 to 1.66 | < 0.001 | 1.37 | 1.32 to 1.42 | < 0.001 |
| **Age group** | | | | | | |
| 18 to 29 | 1.00 | | | 1.00 | | |
| 30 to 39 | 1.74 | 1.57 to 1.94 | < 0.001 | 1.64 | 1.47 to 1.82 | < 0.001 |
| 40 to 49 | 2.99 | 2.71 to 3.30 | | 2.62 | 2.37 to 2.90 | |
| 50 to 59 | 4.47 | 4.07 to 4.92 | | 3.60 | 3.26 to 3.97 | |
| 60+ | 6.09 | 5.58 to 6.66 | | 4.26 | 3.87 to 4.69 | |
| **Ethnic group** | | | | | | |
| White | 1.00 | | | 1.00 | | |
| Non-white | 1.74 | 1.13 to 1.22 | < 0.001 | 1.00 | 0.97 to 1.04 | 0.770 |
| **Education level** | | | | | | |
| 0 to 3 | 1.00 | | | 1.00 | | |
| 4 to 7 | 0.86 | 0.80 to 0.93 | < 0.001 | 0.99 | 0.92 to 1.06 | < 0.001 |
| 8 to 10 | 0.51 | 0.46 to 0.55 | | 0.90 | 0.83 to 0.97 | |
| 11 or more | 0.58 | 0.54 to 0.64 | | 0.94 | 0.86 to 1.02 | |
| **Employed** | | | | | | |
| Yes | 1.00 | | | 1.00 | | |
| No | 1.84 | 1.77 to 1.91 | < 0.001 | 1.20 | 1.15 to 1.26 | < 0.001 |
| **Physical activity** | | | | | | |
| None | 1.00 | | | 1.00 | | |
| Not enough | 1.25 | 1.19 to 1.31 | < 0.001 | 1.03 | 0.98 to 1.08 | 0.350 |
| Enough | 1.45 | 1.39 to 1.50 | | 1.02 | 0.98 to 1.06 | |
| **BMI[d]** | | | | | | |
| Normal weigh | 1.00 | | | 1.00 | | |
| Underweight | 1.06 | 0.90 to 1.23 | < 0.001 | 1.06 | 0.90 to 1.24 | < 0.001 |
| Overweight | 1.37 | 1.31 to 1.43 | | 1.24 | 1.19 to 1.29 | |
| Obesity | 1.77 | 1.69 to 1.85 | | 1.49 | 1.43 to 1.56 | |
| **Alcohol consumption** | | | | | | |
| Abstaining | 1.00 | | | 1.00 | | |
| Moderate | 0.79 | 0.76 to 0.83 | < 0.001 | 0.91 | 0.87 to 0.96 | < 0.001 |
| Excessive | 0.58 | 0.55 to 0.62 | | 0.98 | 0.93 to 1.04 | |
| **Smoking** | | | | | | |
| Never smoked | 1.00 | | | 1.00 | | |
| Ex-smoker | 1.58 | 1.52 to 1.64 | < 0.001 | 1.24 | 1.19 to 1.29 | < 0.001 |
| Smoker | 1.11 | 1.05 to 1.18 | | 1.16 | 1.09 to 1.22 | |
| **Residence area** | | | | | | |
| Urban | 1.00 | | | 1.00 | | |
| Rural | 1.53 | 1.10 to 1.21 | < 0.001 | 1.13 | 0.99 to 1.08 | 0.134 |
| **Brazilian region** | | | | | | |
| North | 1.00 | | | 1.00 | | |
| Northeast | 1.24 | 1.17 to 1.31 | < 0.001 | 1.10 | 1.04 to 1.16 | < 0.001 |
| Southeast | 1.51 | 1.42 to 1.60 | | 1.21 | 1.14 to 1.28 | |
| South | 1.55 | 1.46 to 1.65 | | 1.23 | 1.16 to 1.32 | |
| Midwest | 1.27 | 1.18 to 1.36 | | 1.14 | 1.07 to 1.22 | |
| **Private health insurance** | | | | | | |

(*Continued*)

**Table 2.** (Continued)

| Variables | PR[a] non-adjusted | 95%CI[b] | p[c] | PR adjusted | 95%CI | p |
|---|---|---|---|---|---|---|
| Yes | 1.00 | | | 1.00 | | |
| No | 1.15 | 1.11 to 1.20 | < 0.001 | 1.08 | 1.03 to 1.12 | < 0.001 |
| **Medicines used in the last year** | | | | | | |
| Yes | 1.00 | | | 1.00 | | |
| No | 3.63 | 3.36 to 3.93 | < 0.001 | 2.59 | 2.39 to 2.80 | < 0.001 |

[a]PR: Prevalence ratio

[b]CI: Confidence interval

[c]p: Wald test

[d]BMI: Body mass index.

Individuals with obesity (PR: 1.49; 95%CI: 1.43 to 1.56) and who smoked (PR: 1.16; 95%CI: 1.09 to 1.23) also had a higher prevalence of multimorbidity than individuals with normal weight and those who never smoked, respectively.

The southeast region (PR: 1.23; 95%CI: 1.16 to 1.32) presented the highest prevalence of multimorbidity compared with the other Brazilian regions.

## Discussion

The prevalence of multimorbidity in Brazilian individuals was 29.9%, which was higher than in previous Brazilian surveys (23.6%) [1–4]. Moreover, a high prevalence was found in women (PR: 1.37; 95%CI: 1.32 to 1.42) and individuals over 60 years (PR: 4.26; 95%CI: 3.87 to 4.69), corroborating with worldwide trends for NCDs [5–9,14].

Systemic arterial hypertension, chronic back pain, and hypercholesterolemia were the most prevalent NCDs in both sexes, as observed in NHS 2013 [1]. This result emphasizes the need for attention, especially within the scope of public health.

Regarding associations between sociodemographic factors and multimorbidity, a study conducted with 5,074,227 individuals in the Netherlands observed that unemployed individuals were more likely to have at least three than those employed [1]. These findings corroborated our results since non-employed individuals had a higher prevalence of multimorbidity, regardless of age, sex, or education level.

Individuals with private health insurance had a higher prevalence of multimorbidity than those with no private health insurance. Moreover, individuals who had at least one medical appointment in the year before the survey showed a 2.59-fold increase in the prevalence of multimorbidity compared with those who did not seek a doctor. Access to health services and the accurate diagnosis of NCDs may be easier in private than in public health, favoring prevention. These results may be justified by the question asked during the survey ("Has a doctor ever diagnosed you with (name of the disease)?"), which probably influenced the number of cases of multimorbidity among individuals who had easy access to medical care.

The north region of Brazil, which is the least economically developed, had the lowest prevalence of multimorbidity compared with other regions. In contrast, the southeast and south regions (most economically developed regions) had the highest prevalence of multimorbidity. In addition, the north region of Brazil has specific geographic characteristics (i.e., many rivers, dense vegetation, and a wide territory with low population density) that may justify the difficult access to health services [1,2]. A survey conducted by the Institute of Studies for Health Policies [1] showed that many older individuals from the north region never performed a blood test to measure cholesterol or blood glucose compared with other regions of Brazil. The

literature and our results suggested low access to diagnosis and screening of NCDs in the northern region of Brazil.

Lifestyle factors were significantly associated with multimorbidity. We found a high prevalence of multimorbidity in individuals with obesity (49%) and smokers (24%) compared with those with normal weight and who never smoked, respectively. Similarly, a 20-year cohort study conducted in the United Kingdom with 4,510 individuals aged between 15 and 55 years found an increasing prevalence of multimorbidity. Smoking and obesity were strong independent predictors of multimorbidity [3].

The association between multimorbidity and lifestyle should be a strategy for creating health policies to prevent and treat NCDs in adults and older individuals. These variables may also be important for a comprehensive and multidisciplinary planning, including holistic health and social care, and would reduce the burden of multimorbidity and disadvantages related to lack of prevention, treatment, or follow-up of individuals.

The Brazilian public health system has public policies to treat and control NCDs, such as the Network of Non-Communicable Chronic Diseases, the National Program for Attention to Arterial Hypertension and Diabetes Mellitus, and many other programs created especially for women and older individuals. Based on this context and the correlation with other diseases, it is possible to treat or control NCDs using lifestyle and therapeutic interventions, justifying the public investment in these policies [1–4].

The present study had some limitations. Information on self-reported chronic diseases may be less accurate than objective data and are subject to recall bias. Furthermore, access to health services, or lack thereof, may have contributed to underestimating the prevalence of multimorbidity. Another limitation of this study concerns the household sample that excluded institutionalized elderly, indigenous and quilombola groups, among others, which could further contribute to a reduced prevalence of NCDs and access to health services. However, all these biases were considered, so the use of a population database and complex statistical analyzes allowed us to extrapolate the results to the entire Brazilian population.

## Conclusion

The results indicate a high prevalence of multimorbidity in Brazilian adults, estimated at 29.9%, which would correspond to 47,598,091 individuals. The most frequently observed conditions were systemic arterial hypertension, chronic back pain, and hypercholesterolemia. Multimorbidity was associated with sociodemographic and lifestyle characteristics and is more common in women and the elderly.

Understanding the association between different health conditions and the development of these diseases makes it possible to create programs and services aligned with the needs of the population in different social conditions and stages of life.

## Author Contributions

**Conceptualization:** Ana Daniela Silva da Silveira, Jonas Eduardo Monteiro dos Santos, Marianna de Camargo Cancela, Dyego Leandro Bezerra de Souza.

**Data curation:** Ana Daniela Silva da Silveira, Jonas Eduardo Monteiro dos Santos, Marianna de Camargo Cancela, Dyego Leandro Bezerra de Souza.

**Formal analysis:** Ana Daniela Silva da Silveira, Jonas Eduardo Monteiro dos Santos, Marianna de Camargo Cancela, Dyego Leandro Bezerra de Souza.

**Funding acquisition:** Ana Daniela Silva da Silveira, Jonas Eduardo Monteiro dos Santos, Marianna de Camargo Cancela, Dyego Leandro Bezerra de Souza.

**Investigation:** Ana Daniela Silva da Silveira, Jonas Eduardo Monteiro dos Santos, Marianna de Camargo Cancela, Dyego Leandro Bezerra de Souza.

**Methodology:** Ana Daniela Silva da Silveira, Jonas Eduardo Monteiro dos Santos, Marianna de Camargo Cancela, Dyego Leandro Bezerra de Souza.

**Project administration:** Ana Daniela Silva da Silveira, Jonas Eduardo Monteiro dos Santos, Marianna de Camargo Cancela, Dyego Leandro Bezerra de Souza.

**Resources:** Ana Daniela Silva da Silveira, Jonas Eduardo Monteiro dos Santos, Marianna de Camargo Cancela, Dyego Leandro Bezerra de Souza.

**Software:** Ana Daniela Silva da Silveira, Jonas Eduardo Monteiro dos Santos, Marianna de Camargo Cancela, Dyego Leandro Bezerra de Souza.

**Supervision:** Ana Daniela Silva da Silveira, Jonas Eduardo Monteiro dos Santos, Marianna de Camargo Cancela, Dyego Leandro Bezerra de Souza.

**Validation:** Ana Daniela Silva da Silveira, Jonas Eduardo Monteiro dos Santos, Marianna de Camargo Cancela, Dyego Leandro Bezerra de Souza.

**Visualization:** Ana Daniela Silva da Silveira, Jonas Eduardo Monteiro dos Santos, Marianna de Camargo Cancela, Dyego Leandro Bezerra de Souza.

**Writing – original draft:** Ana Daniela Silva da Silveira, Jonas Eduardo Monteiro dos Santos, Marianna de Camargo Cancela, Dyego Leandro Bezerra de Souza.

**Writing – review & editing:** Ana Daniela Silva da Silveira, Jonas Eduardo Monteiro dos Santos, Marianna de Camargo Cancela, Dyego Leandro Bezerra de Souza.

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
