## [Decision Letter · Decision Letter 0]

15 Mar 2023

PONE-D-22-31513Prevalence of multimorbidity in Brazilian individuals: a population-based studyPLOS ONE

Dear Dr. Silva da Silveira,

Thank you for submitting your manuscript to PLOS ONE. After careful consideration, we feel that it has merit but does not fully meet PLOS ONE’s publication criteria as it currently stands. Therefore, we invite you to submit a revised version of the manuscript that addresses the points raised during the review process.

The manuscript has been reassessed by two reviewers. Their comments are appended below. The reviewers observe some major concerns about the manuscript, and in particular they feel that important methodological issues exist that affect the technical soundness of your study, and the conclusions of the paper. We request authors to conduct a detailed review of the comments.

We look forward to receiving your revised manuscript.

Kind regards,

Bruno Pereira Nunes, Ph.D.

Academic Editor

PLOS ONE

Journal Requirements:

Reviewers' comments:

Reviewer's Responses to Questions

**Comments to the Author**

1. Is the manuscript technically sound, and do the data support the conclusions?

Reviewer #1: Partly

Reviewer #2: No

2. Has the statistical analysis been performed appropriately and rigorously? 

Reviewer #1: I Don't Know

Reviewer #2: No

3. Have the authors made all data underlying the findings in their manuscript fully available?

Reviewer #1: Yes

Reviewer #2: No

4. Is the manuscript presented in an intelligible fashion and written in standard English?

Reviewer #1: No

Reviewer #2: Yes

5. Review Comments to the Author

Reviewer #1: In the Methodology Section please provide the following :

• Basic description of multistage sampling used and clarification at what level(s) was the National Health Survey (NHS) representative;

• Detailed description of all ethical considerations of the NHS;

• Explanation on how NHS data became public with unrestricted access?

• Explanation why did you analyze 88,531 responses and why multimorbidity was defined by the presence of two or more NCDs and later in the text you use the term condition?

• Description of all independent variables used (for example: level of education, what do working status and minutes of physical activity mean);

• Description of prerequisites for and procedure of the Poisson regression used.

For the Results Section please do the following:

• As Prevalence Ratio (PR) and Odds Ratio (ORs) are not the same thing and logistic regression coefficient B, Exp(B) and Wald test have values with regard to p value (Sig.), perform analyses again and use clearly defined statistical marks in the presenting tables.

Reviewer #2: The manuscript addresses an emerging theme in the literature. However, the rationale is very superficial. The methods need more detail and the adjusted analysis seems to me to have many flaws, there is a need to review the adjustments. There is no focus on evaluating MM associated with any specific factor and this makes the manuscript much more superficial in other points, such as the discussion that needs greater depth. Comments were made in the attached text.

6. PLOS authors have the option to publish the peer review history of their article (what does this mean?). If published, this will include your full peer review and any attached files.

Reviewer #1: No

Reviewer #2: No

---

## [Author Response · Author response to Decision Letter 0]

18 Apr 2023

Dear reviewers, all your recommendations have been accepted. All changes were described in the documents attached on the journal's platform. We appreciate the suggestions. The authors.

---

## [Decision Letter · Decision Letter 1]

25 May 2023

PONE-D-22-31513R1Prevalence of multimorbidity in Brazilian individuals: a population-based studyPLOS ONE

Dear Dr. Silva da Silveira,

Thank you for submitting your manuscript to PLOS ONE. After careful consideration, we feel that it has merit but does not fully meet PLOS ONE’s publication criteria as it currently stands. Therefore, we invite you to submit a revised version of the manuscript that addresses the points raised during the review process.

The manuscript has been reassessed by two reviewers. Their comments are appended below. The reviewers continue to observe major concerns about the manuscript, and in particular they feel that important methodological issues exist that affect the technical soundness of your study, and the conclusions of the paper. We request authors to conduct a detailed review of the comments.

We look forward to receiving your revised manuscript.

Kind regards,

Bruno Pereira Nunes, Ph.D.

Academic Editor

PLOS ONE

Reviewers' comments:

Reviewer's Responses to Questions

**Comments to the Author**

1. If the authors have adequately addressed your comments raised in a previous round of review and you feel that this manuscript is now acceptable for publication, you may indicate that here to bypass the “Comments to the Author” section, enter your conflict of interest statement in the “Confidential to Editor” section, and submit your "Accept" recommendation.

Reviewer #1: (No Response)

Reviewer #2: All comments have been addressed

2. Is the manuscript technically sound, and do the data support the conclusions?

Reviewer #1: Partly

Reviewer #2: Partly

3. Has the statistical analysis been performed appropriately and rigorously? 

Reviewer #1: I Don't Know

Reviewer #2: No

4. Have the authors made all data underlying the findings in their manuscript fully available?

Reviewer #1: No

Reviewer #2: Yes

5. Is the manuscript presented in an intelligible fashion and written in standard English?

Reviewer #1: Yes

Reviewer #2: Yes

6. Review Comments to the Author

Reviewer #1: Please provide in the Methodology section:

1. what is the total number of participants covered by the analyses?

2. description why and how did you use Incidence Rate Ratio (IRR),

3. proofs of assumptions for the use of Poisson regression,

4. detailed procedure for the use of Poisson regression.

Please provide in the Results section correct interpretations for PR (in comparison to what it is high / low)

Reviewer #2: The rationale of the study does not seem to me sufficient. There is already recent evidence of MM and associated factors, including a longer list of PNS morbidities. The authors also need to explain the analysis model used, it is not clear. The list of diseases must be described and not just mention the figure. Provide an explanation for these variables (measurements):physical activity, body mass index (BMI), alcohol consumption, smoking, private health insurance, and medical care in the last year. In the discussion, the mention "memory bias" should be modified. I believe it is recall bias. Furthermore, limitations should be discussed and not just cited.

7. PLOS authors have the option to publish the peer review history of their article (what does this mean?). If published, this will include your full peer review and any attached files.

Reviewer #1: No

Reviewer #2: No

---

## [Author Response · Author response to Decision Letter 1]

11 Oct 2023

We appreciate the new suggestions for publication. The authors carefully reviewed the manuscript "Prevalence of multimorbidity in Brazilian individuals: a population-based study". Our responses to the considerations of Reviewer n#1 and Reviewer n#2 are listed below

Dear Reviewer n#1, we accepted your suggestions and made the necessary adjustments according to your guidelines. We are grateful for that and have provided the answers to your questions below:

The Reviewer #1 says:

"Please find below my remarks:

1. On the Page 4, Line 78 – Authors write: “We analyzed 88,531 responses about the lifestyle of Brazilian individuals corresponding to the responses of people over 18 years old.”; and,

2. on the Page 6, Line 117 authors write: “The NHS 2019 involved 88,531 individuals corresponding to an estimated 159,171,311 individuals”; and,

3. present on the Page 7, Table 1. Estimated prevalence of multimorbidity according to sociodemographic characteristics, social determinants, lifestyle factors, and use of health services, with the number of participants of 47,598,091; and, 

4. write statement, as an answer to the reviewer, that “In this study, we chose to carry out the Poisson regression model, the estimated measure used was the IRR (Incidence Rate Ratio) and the Wald test was used as post-estimation of the Poisson model. We do not use logistic regression”."

A: All suggestions accepted and text corrected as explained below

The Reviewer #1 ask:

1. what is the number of participants covered by the authors analyses?

A: The number of responses analyzed in this research was 88,531 (On page 4, line 78 and page 6, line 117), corresponding to responses from individuals aged over 18 years. The SVY module for complex statistics made it possible to estimate these values for the total number of responses (159,171,311; page 6, line 117) and, of this total, the number of people in the category "with multiborbity" (people with 2 or more chronic diseases) is 47,598,091, represented in Table 1 (Page 7, Table 1).

Reviewer #1 - 2&3: 

• write statement, as an answer to the reviewer, that “In this study, we chose to carry out the Poisson regression model, the estimated measure used was the IRR (Incidence Rate Ratio) and the Wald test was used as post-estimation of the Poisson model. We do not use logistic regression”. and

2. why and how did authors use Incidence Rate Ratio (IRR) for data from typical prevalence study such as National Health Survey? (an incidence rate ratio allows us to compare the incident rate between two different groups, exposed vs. non-exposed). and

3. where did they present assumptions for the use of Poisson regression? (two out of four assumptions should be that the distribution of counts follows a Poisson distribution and mean and variance of the model are equal)

A: The classic Poisson probability distribution is applied to count data to model rates over time and predict whether the mean and variance are equal. However, in situations of underdispersion or overdispersion of the data – in which the mean and variance are not equal – it is possible to make adaptations to the classical model, using link functions. In this case, using Poisson modeling fits the binary result data using the logarithmic function to linearize the relationship between the result and the independent variables. It's the same assumption as logistic regression, but uses the logit link function. In the classic use of the Poisson distribution, the measure of association, after beta exponentiation, is the IRR, like the incidence rate of a given event. However, in the linearization process with the logarithmic function, the classic association measure can be interpreted as a prevalence ratio, as it is cross-sectional data.

Reviewer #1 - 4: 

4. why they interpret obtained ratios as high not giving belonging comparison (for example if IRR Greater than 1: This indicates that the incident rate is greater in an exposed group compared to an unexposed group). 

A: Suggestion accepted and text corrected (Page 8, Line 127 and 135)

Dear Reviewer n#2, we accepted your suggestions and made the necessary adjustments according to your guidelines. We are grateful for that.

---

## [Decision Letter · Decision Letter 2]

14 Nov 2023

PONE-D-22-31513R2Prevalência de multimorbidade em indivíduos brasileiros: um estudo de base populacionalPLOS ONE

Dear Dr. Silva da Silveira,

Thank you for submitting your manuscript to PLOS ONE. After careful consideration, we feel that it has merit but does not fully meet PLOS ONE’s publication criteria as it currently stands. Therefore, we invite you to submit a revised version of the manuscript that addresses the points raised during the review process.

We look forward to receiving your revised manuscript.

Kind regards,

Bruno Pereira Nunes, Ph.D.

Academic Editor

PLOS ONE

**Additional Editor Comments:**

The manuscript has undergone a thorough reassessment by one designated reviewer, and their comments are provided below for your consideration. Some concerns have been raised regarding the manuscript's rationale and methodological aspects, which impact the overall technical robustness of your study. We kindly urge the authors to carefully review the provided comments and incorporate necessary revisions into the text.

Reviewers' comments:

Reviewer's Responses to Questions

**Comments to the Author**

1. If the authors have adequately addressed your comments raised in a previous round of review and you feel that this manuscript is now acceptable for publication, you may indicate that here to bypass the “Comments to the Author” section, enter your conflict of interest statement in the “Confidential to Editor” section, and submit your "Accept" recommendation.

Reviewer #2: (No Response)

2. Is the manuscript technically sound, and do the data support the conclusions?

Reviewer #2: Partly

3. Has the statistical analysis been performed appropriately and rigorously? 

Reviewer #2: No

4. Have the authors made all data underlying the findings in their manuscript fully available?

Reviewer #2: No

5. Is the manuscript presented in an intelligible fashion and written in standard English?

Reviewer #2: No

6. Review Comments to the Author

Reviewer #2: (No Response)

7. PLOS authors have the option to publish the peer review history of their article (what does this mean?). If published, this will include your full peer review and any attached files.

Reviewer #2: No

---

## [Author Response · Author response to Decision Letter 2]

2 Dec 2023

The last revision came without any files. After contacting the PLOS ONE editorial committee, I am resending the corrections to the latest version, which follows:

Dear Reviewer n#1, we accepted your suggestions and made the necessary adjustments according to your guidelines. We are grateful for that and have provided the answers to your questions below:

The Reviewer #1 says:

"Please find below my remarks:

1. On the Page 4, Line 78 – Authors write: “We analyzed 88,531 responses about the lifestyle of Brazilian individuals corresponding to the responses of people over 18 years old.”; and,

2. on the Page 6, Line 117 authors write: “The NHS 2019 involved 88,531 individuals corresponding to an estimated 159,171,311 individuals”; and,

3. present on the Page 7, Table 1. Estimated prevalence of multimorbidity according to sociodemographic characteristics, social determinants, lifestyle factors, and use of health services, with the number of participants of 47,598,091; and, 

4. write statement, as an answer to the reviewer, that “In this study, we chose to carry out the Poisson regression model, the estimated measure used was the IRR (Incidence Rate Ratio) and the Wald test was used as post-estimation of the Poisson model. We do not use logistic regression”."

A: All suggestions accepted and text corrected as explained below

The Reviewer #1 ask:

1. what is the number of participants covered by the authors analyses?

A: The number of responses analyzed in this research was 88,531 (On page 4, line 78 and page 6, line 117), corresponding to responses from individuals aged over 18 years. The SVY module for complex statistics made it possible to estimate these values for the total number of responses (159,171,311; page 6, line 117) and, of this total, the number of people in the category "with multiborbity" (people with 2 or more chronic diseases) is 47,598,091, represented in Table 1 (Page 7, Table 1).

Reviewer #1 - 2&3: 

• write statement, as an answer to the reviewer, that “In this study, we chose to carry out the Poisson regression model, the estimated measure used was the IRR (Incidence Rate Ratio) and the Wald test was used as post-estimation of the Poisson model. We do not use logistic regression”. and

2. why and how did authors use Incidence Rate Ratio (IRR) for data from typical prevalence study such as National Health Survey? (an incidence rate ratio allows us to compare the incident rate between two different groups, exposed vs. non-exposed). and

3. where did they present assumptions for the use of Poisson regression? (two out of four assumptions should be that the distribution of counts follows a Poisson distribution and mean and variance of the model are equal)

A: The classic Poisson probability distribution is applied to count data to model rates over time and predict whether the mean and variance are equal. However, in situations of underdispersion or overdispersion of the data – in which the mean and variance are not equal – it is possible to make adaptations to the classical model, using link functions. In this case, using Poisson modeling fits the binary result data using the logarithmic function to linearize the relationship between the result and the independent variables. It's the same assumption as logistic regression, but uses the logit link function. In the classic use of the Poisson distribution, the measure of association, after beta exponentiation, is the IRR, like the incidence rate of a given event. However, in the linearization process with the logarithmic function, the classic association measure can be interpreted as a prevalence ratio, as it is cross-sectional data.

Reviewer #1 - 4: 

4. why they interpret obtained ratios as high not giving belonging comparison (for example if IRR Greater than 1: This indicates that the incident rate is greater in an exposed group compared to an unexposed group). 

A: Suggestion accepted and text corrected (Page 8, Line 127 and 135)

Dear Reviewer n#2, we accepted your suggestions and made the necessary adjustments according to your guidelines. We are grateful for that.

---

## [Editor Report · Decision Letter 3]

12 Dec 2023

Prevalence of multimorbidity in Brazilian individuals: a population-based study

PONE-D-22-31513R3

Dear Dr. Silva da Silveira,

We’re pleased to inform you that your manuscript has been judged scientifically suitable for publication and will be formally accepted for publication once it meets all outstanding technical requirements. **Additionally, it is necessary for you to make the following adjustments for the final version of the manuscript:**

1) Line 105: review and replace (if deemed appropriate) "robust variance" with robust adjustment of variance;

2) Line 209: is it memory bias or recall bias?;

3) review the English throughout the text;

4) send figures with greater clarity and resolution.

Kind regards,

Bruno Pereira Nunes, Ph.D.

Academic Editor

PLOS ONE
---

## [Editor Report · Acceptance letter]

19 Dec 2023

PONE-D-22-31513R3 

PLOS ONE

Dear Dr. Silva da Silveira, 

I'm pleased to inform you that your manuscript has been deemed suitable for publication in PLOS ONE. Congratulations! Your manuscript is now being handed over to our production team.

Kind regards, 

on behalf of

Dr. Bruno Pereira Nunes 

Academic Editor

PLOS ONE